# A Qualitative Exploration of Factors Influencing Non-Use of Sexual Reproductive Health Services among University Students in South Africa

**DOI:** 10.3390/ijerph20032418

**Published:** 2023-01-29

**Authors:** Ntombenhle E. Mazibuko, Munyaradzi Saruchera, Emeka Francis Okonji

**Affiliations:** 1Faculty of Economic and Management Sciences, Stellenbosch University, Stellenbosch 7599, South Africa; 2School of Public Health, University of the Western Cape, Cape Town 7535, South Africa

**Keywords:** sexual reproductive health, psychosocial support, HIV, AIDS, contraceptives, condoms

## Abstract

(1) Background: There is growing concern in South Africa about risky sexual behaviour, sexual transmitted infections (STIs), and unplanned pregnancy among young people. Many sexually active students engage in several risky behaviours, including sex with multiple sexual partners, low condom use, and low contraceptive use. This paper qualitatively explores factors influencing non-use of sexual reproductive health services by students at Mangosuthu University of Technology in South Africa (MUT). (2) Methods: Data was collected through 20 in-depth interviews with MUT students and subjected to inductive thematic analysis. Informed consent was obtained before all data collection. (3) Results: The main themes identified were risky sexual behaviours translating to multiple intimate partners, perceived quality of condom use, perceived benefits of contraceptives, negotiating safer sex with partners, developing a greater sense of autonomy, alcohol and drug abuse, perceived benefits of health education provided by the MUT, and lack of open communication. (4) Conclusions: The findings suggest that university students need multi-faceted interventions designed to address challenges with risky sexual behaviours including knowledge and benefits of condom and contraceptive use to prevent STIs and unwanted pregnancies, as well as providing psychosocial interventions to support these students’ autonomy.

## 1. Introduction

Young people appear to be involved in risky sexual behaviours (e.g., early sexual debut, multiple sexual relationships and non-condom use). This makes them vulnerable to numerous health issues such as sexually transmitted infections (STIs), human immunodeficiency virus (HIV) and acquired immunodeficiency syndrome (AIDS). Many studies have shown that high-risk sexual behaviours promote the spread of HIV, STIs and unplanned pregnancies among young people [1]. University students make up a significant proportion of young people and sexually active people who are at a higher risk of acquiring HIV [2]. The academic community creates an intervention framework for high-risk sexual behaviours, particularly condomless sexual relationships and multiple sexual partnerships [3,4,5,6], which increase the risk of contracting HIV and STIs. Although students are well aware of the risks associated with unsafe sexual behaviours, many students continue to engage in risky behaviours including having multiple sexual partners and unprotected sexual intercourse [6]. 

In several low-and middle-income countries (LMICs) including South Africa, there is evidence of early sexual debuts among young people [7]. Data from a cohort study conducted in South Africa indicate that the median age of sexual debut in South Africa is 15 years for males and 16 years for females [7]. This early sexual activity has the potential to increase the risk of unintended pregnancies, HIV, and STIs. With early sexual activity onset in young people, there is serious concern about the health implications of such an early sexual initiation. The harmful consequences of unprotected sex in young people lead to incurable STIs, HIV infection and unplanned pregnancies [8]. It is important to promote knowledge about responsible sex behaviours before young people become sexually active. Risk reduction measures such as abstinence from sexual intercourse, delayed sexual initiation, reduced number of sexual partners and increased access to and use of condoms is important in preventing young people from acquiring STIs, as well as HIV and unwanted pregnancies [3].

There is growing concern in South Africa about risky sexual behaviours, STIs, and unplanned pregnancies among young people. Many sexually active students engage in several risky behaviours, including sex with multiple sexual partners, low condom use, and low contraceptive use [6]. Such behaviours make them vulnerable to STIs/HIV infection and unplanned pregnancies. Although several efforts have been put in place to reduce the risk of sexual behaviours of young people using different strategies, policies and activities at the national level, more interventions are needed to reduce risky sexual behaviours among young people. Therefore, this paper’s aim is to understand the factors driving the high incidence of unplanned pregnancy and STI/HIV infection among students at Mangosuthu University of Technology (MUT). Research exploring the factors influencing STIs, HIV infection and unplanned pregnancy among young people at the university despite the provision of sexual and reproductive health services at the university can help inform the prevention and education programmes. 

## 2. Materials and Methods

### 2.1. Study Setting

MUT is a South African University of Technology. The student population is made up of males, females and others in the age group of 17–30 years. Students enrolled at MUT live in the institution’s residences, both on and off campus, while others live in private accommodations, such as rental houses and those who live in their own homes. As of mid-2021, there were 14,590 students enrolled at MUT, including 14,546 Black African students, two White students, sixteen Indian students, and twenty-six students of colour (MUT Student Registration Office, 2021). Most students are from South Africa, and a few from neighbouring countries; for example, eight from Zimbabwe, ten from Eswatini, one from Lesotho, and two from the Democratic Republic of Congo (DRC) (MUT Student Registration Office, 2021).

### 2.2. Study Design, Sampling, and Data Collection

We applied a qualitative descriptive research design. During one of the routine MUT health awareness events conducted via Teams, held from 14–25 June 2021, students were introduced and recruited for the study. All students (589) who attended the health awareness event had an equal opportunity to participate in the study and were given contact information for research assistants to contact if they wished to participate in the study. The students provided their email addresses, which were used to invite them for interviews and to set up appointments. In the email list of potential research participants, the researcher started with the first address listed at the top of the first page of the list and followed the list to the last email address. Between July and September 2021, 20 one on one in-depth interviews were conducted via Teams due to COVID-19 restrictions and protocols. The interviews were conducted with undergraduate and post-graduate students in MUT who were in their first, second, and third years of study as well as the postgraduates. 

The interview guides were developed in English and piloted for clarity. Interviews were conducted in English via Microsoft Teams and facilitated by a researcher. The interviews were recorded using Microsoft Teams recording, as well as a separate audio recording device. A notebook was also used. The research assistant facilitated the coordination of the recruitment and advertisement of the study during the awareness event. The interviews were audio-recorded and transcribed by the researcher who facilitated that interview. The main aspects explored were (i) participants’ use of condoms and contraceptives, (ii) participants’ views on health education at MUT and condom and contraceptive use, (iii) barriers to condom and contraceptive use, and (iv) participants’ recommendations for improving health services provided by the MUT Clinic.

### 2.3. Inclusion and Exclusion Criteria

The inclusion criteria for the study were as follows:

MUT full-time students; male, and female and others; first to third year and postgraduate students. All participants in the study reported being either male or female. The reason for including all gender identities was to ensure representativeness and to obtain a fair mix of views on personal responsibility and the different health needs and services of all groups, and campus residential and non-campus residential students. 

Participants were excluded if: 

They were employees and contractors of MUT because they were not involved with the identified health problems prevalent at MUT. Table A1 shows the descriptive characteristics of the participants included in the study.

### 2.4. Data Analysis

Two researchers coded and transcribed the data from the in-depth interviews, which were analysed using an iterative process. An initial codebook was developed deductively from the interview guide and inductively from the transcripts. The inductive content analysis technique conducted is suitable for analysing qualitative data collected without the guidance of a theoretical framework but deriving purely from emergent data and formulating themes [9]. The data were captured in an Excel table with five columns in the order of categories, themes, subthemes, codes and quotations or remarks (Table 1). We developed codes from the responses provided by the participants. The data codes were reviewed by the authors and later organised into subthemes, themes, and categories (pattern coding). We further read and cross-checked the transcribed data to confirm the developed pattern coding. Data saturation was achieved where no new themes and subthemes emerged from the in-depth interviews at the individual level.

### 2.5. Trustworthiness and Credibility

The trustworthiness and credibility of the study were ascertained by performing the following. First, we piloted the interview questionnaires with participants to ensure that the questions were relevant and sensitive to solicit a response. Second, we conducted iterative questioning in data collection dialogues with each study participant. The participants were assured that there were no wrong answers, as their experiences were paramount. Finally, we followed the relevant aspects of the consolidated criteria for reporting qualitative research (COREQ) outlined by Tong et al. [10,11]. The COREQ checklist aimed to promote clear and comprehensive reporting guidelines for qualitative studies.

### 2.6. Ethics Approval and Informed Consent

The researchers sought verbal and written consent from all study participants. The researchers received ethical clearance from the MUT Research Council, the MUT Campus Health Clinic Head of Department (HOD) and from Stellenbosch University. Pseudonyms were used to identify study participants in the transcription of the in-depth interviews. In addition, the participants consented to the publishing of their responses if their identities were kept anonymous. No personal information was collected from participants during the research process. 

## 3. Results

Several themes emerged from the interviews, which were grouped into different codes and themes for analysis (Table 1). The main themes identified were risky sexual behaviours, translating to multiple intimate partners, perceived quality of condom use, perceived benefits of contraceptives, developing a greater sense of autonomy, alcohol and drug abuse, and perceived benefits of the health education provided by the MUT. 

**Table 1 ijerph-20-02418-t001:** Themes, subthemes, and codes inductively extracted from the transcripts.

Category	Theme	Sub Theme	Codes	Quotations
Sexual and reproductive health services	Multiple intimate partner		Intimate partners	I have one sexual partner
				Yes, I do, I have 2 sexual partners, because I am still studying the minds of boys. 21-year-old female
				One sex partner now. Had multiple partners before-21 years old
				Multiple sexual partners, with three females. Two partners know each other, and they are comfortable with it. The possibility of having another partner when she is with me is there. I am comfortable with that. I am in an open relationship.
				I have no sex partner since the beginning of 2021 hence I am not sexually active, but I have had one before.
	Have no problem with condom quality	Comfortable with supplied condoms	Happy with supplied condoms	I sometimes use supplied condoms from MUT clinic
				We both are comfortable. We both do not want a child now. I am already suffering with
				I also buy condoms from the pharmacy- 27 years old female
			Abstaining from sex	[I am currently] abstaining from sex
				[I] stopped sex some time ago
		Inconsistent use of condoms		I use condoms sometimes. I use it when I have sex with my other sex partner and not the main guy
				No, I use condoms with two of my partners all the time and not with the third one. I have my personal reasons for that- 22 years old male
				In fact, all condoms are uncomfortable truly, to you and your partner. It feels dry and the partner sometimes just pulls it off due to discomfort. It just feels ok to enjoy the intimacy knowing that you did not put it on and just exploring that moment, the feeling and the sensation. The thing is that we should be using the condom all the time, but we are not
				We use condoms most of the time. Sometimes we did not use condoms because we did not have them
			Misconception of condom use	The only time I did not use condoms is when I was breaking her virginity. After that, I was always using it until we broke up.”
			Familiarity with each other	I will not say we use 100%, sometimes. We have been together for a long time
				I am not using condoms all the time, only with the partner I see for the first time. I am aware of STIs and HIV infection and that is why I always use condoms with non-regular sexual partners. I had a condom burst twice and I had STIs three times, so I am scared of these disease
		Using condoms consistently for fear of pregnancy and sexually transmitted infections		I always used the condoms. I was scared I could impregnate her if I did not use the condoms. That is why I always used condoms. No condom burst, No STIs. I know my HIV status as well as of my ex-girlfriend.
				I had an unplanned pregnancy because we were not using condoms all the time but since I fell pregnant, I make sure I always have condoms even in my bag even if I know I am not going to have any sex.
			Knowledge of condom use	[I] knows enough of the use of condom say I am about 70 percent; I need to learn more
	Perceived quality of free condoms distributed	Prefers bought condoms		I prefer the ones I buy because some of us do not like the scented condoms like the one that are supplied
				Prefer Icon condoms. It has good lubricant, and it is strong, because they have bigger size as compared to other condoms types.
				I prefer the ones that I buy. I have had condom break twice with the supplied condoms but the ones that I bought have never disappointed me [not] even once. Durex is nicer, it feels like almost not there
			Feels like having sex without condoms	I prefer the condoms that I buy in terms of quality and comfort. They feel like skin on skin as compared to the ones that are supplied by the MUT Clinic. My partner feels the same. Durex feather light condoms is like that
	Perceived benefits of contraceptive use	Using contraceptives	Knowledge of contraceptive use	Yes, the two partners are on contraceptives but the third one is not on them, and we do not use condoms. We are ready for pregnancy if it happens. Financially we are ready as well-22 years male
				The third one is on contraceptives but the first two; I think they are not on them; I am not sure. They would have told me
				I am on Petogen
			Consistently using contraceptives	Yes, I am on Nur-Isterate, and I am ok with it for almost 2 years now with no problems.
				I am on Implanon which I got at MUT Clinic.
				No, she is not using contraceptives, she did tell me that she used them before I came to her life, but she stopped now. I do not know why she stopped them-23 years old male
				I am on Implanon which I got at MUT Clinic.
				Uses the pill from MUT
			Non use of contraceptives	No contraceptives, only condom
				*“… you gain weight; people get pregnant while using contraceptives like Implant…”*-30-year-old female.
				I did not know if she was on contraceptives. We never discussed that.
			Inconsistent use of contraceptives	Missed pill dates
				I used to miss out on my return date for my injection.
			Contraceptive awareness	I need to be safe instead of being reckless.
				I was screened and I kept going to the clinic to do pregnancy check and STIs screening
				“What I told myself is that at least let me have a first baby then afterwards, maybe I can use contraceptives because I cannot prevent without having any baby. I do not have a baby. That is a bit risky for my health and my womb”- 21-year- old female
Social challenges	Able to negotiate safer sex practices		Knows and practice her rights on condom use and contraceptives	Yes, I know, and I do practice my rights. My partner was ok with it.
			Use condoms because of fear of contracting diseases	We need to use condoms because we do not stay together, and we do not know what each is doing when we are not together.
	Developing a greater sense of autonomy		Newfound freedom	I see a lot of students who are pregnant, and I see it as a problem. I think this is due to students being uneducated about unprotected sex and how dangerous it is. These children are still young from high school. They need to be taught about these problems. I think this is because they have just found freedom and they fall into trap
	Alcohol and drug abuse	Alcohol and drug abuse	No condom use when drunk	There will be no condom use if both partners are drunk because they will not be thinking straight. They will want to do it now and their judgement will be disturbed.
				No condom use when drunk. Drinks alcohol, smokes Cannabis and cigarettes
				Much as some people may not want to admit it, the truth is when you are drunk, the normal you and the current you are not the same. It will be hard to put in good judgement. Decisions are just made without a second thought. Some people drink until they blackout. Hormones also have an effect. Other people want sex more and are uninhibited when they are drunk. They end up doing things they would not have done if they were not drunk. I am sure that is when most people have unprotected sex, when they are drunk
				Alcohol does not change me. If I planned to use condom, I will use it and if I do not want, I will not use it irrespective of how drunk I am.
	Perceived benefits of health education provided by MUT			“I would not say yes because it’s things that I have learnt before from my life orientation classes in high school. It is just the reminders. I have not attended any health campaign; I knew about them, but it never came to me to attend them.”
				“Yes, judging from what is happening, it is not working. You know us, students, we do not listen. Maybe it is the culture with us; maybe it is the times that we are growing in. If someone says, do not touch this fire, you will be burnt, we want to touch it and see if we will be burnt for real

### 3.1. Risky Sexual Behaviour

In general, most students are quite familiar with the term “risky sexual behaviour”. The students were aware of the behaviours that expose them to increased risks of pregnancy and STIs. The students identified the risks associated with unprotected sex. When the students referred to unprotected sex, most made reference to penetrating vaginal intercourse. The students also identified intercourse with multiple sex partners as risky behaviour.

Six out of the twenty participants reported having multiple sexual partners. Five males had two or three intimate partners, while one was female. When asked if she had multiple intimate partners, the participant candidly answered:

“Yes I do, I have two partners.” Twenty-one-year-old female.

The female participant stated that she had multiple partners for fun because she was “still studying the minds of boys.” She was just fooling around and studying boys’ minds.

One male participant had two sexual partners, but stopped before the time of the study.

However, there were differences between male and female students’ perception on risky sexual behaviours. More males reported sexual activity compared to females. Only one student revealed that he had never had sex previously. Two participants were sexually active previously but were no longer sexually active at the time of the study.

### 3.2. Perceived Quality of Condoms That Are Freely Distributed

Consistent condom use is one of the most effective strategies to reduce health risks associated with STIs and pregnancies. The study found varying levels of condom use and perceived attitudes towards the quality of free condoms distributed at MUT. Of the 17 participants who were sexually active at the time of the study, 12 participants reported using condoms, and four reported using condoms consistently. Only one participant did not use condoms at all. Most participants (17 out of 20) preferred condoms that they purchased themselves as opposed to those freely supplied by the MUT Clinic. When asked the reasons for their preference, one of the respondents stated.

“I prefer the ones that I buy. I have had condom break twice with the supplied condoms but the ones that I bought have never disappointed me [not] even once. Durex is nicer, it feels like almost not there.”

Another participant ascribed the quality of purchased condoms to the feeling of skin-to-skin contact.

“I prefer the condoms that I buy in terms of quality and comfort. They feel like skin on skin as compared to the ones that are supplied by the MUT Clinic. My partner feels the same.”

Additionally, another participant spoke about his dislike for scented condoms, which is common with distributed condoms.

“I prefer the ones I buy because some of us do not like the scented condoms like the one that are supplied”.

Though most of the participants reported using condoms, preferences for purchased condoms abound due to the reasons highlighted above. However, few participants continue to use the freely distributed condoms. The participants who reported using condoms expressed the fear of getting pregnant as the motivation for consistent use of condoms.

“I had an unplanned pregnancy because we were not using condoms all the time but since I fell pregnant, I make sure I always have condoms even in my bag even if I know I am not going to have any sex”.

This finding of the study suggests that slightly more than half (13) of the participants surveyed had experienced STIs and HIV infections as well as unplanned pregnancies, which may contribute to and explain the high rates of STIs and HIV infections as well as unplanned pregnancies and abortions as a result of inconsistent use of freely distributed condoms and contraceptives.

One of the participants who had multiple (two) intimate partners justified the inconsistent condom use by saying that she trusted her main partner, even though she herself had multiple sexual partners.

“I use condoms sometimes. I use them when I have sex with my other sex partner and not the main guy. I have two sex partners.”

The study found that some students used condoms only in the early stages of their relationship. One of the study participants reported using condoms only when he first saw his partner or with an irregular partner. Respondent eight also had multiple partners and stated:

“I am not using condoms all the time, only with the partner I see for the first time. I am aware of STIs and HIV infection and that is why I always use condoms with non-regular sexual partners. I had a condom burst twice and I had STIs three times, so I am scared of these diseases.”

Another participant (participant three) said that all condoms were uncomfortable:

“In fact, all condoms are uncomfortable, truly to you and your partner it feels dry, and the partner sometimes just pulls it off due to discomfort. It just feels ok and enjoys the intimacy knowing that you did not put it on and just exploring that moment, the feeling and the sensation. The thing is that we should be using the condom all the time, but we are not.”

Three of the 20 participants demonstrated some level of misinformation or lack of knowledge about contraceptives and STI prevention. One participant stated that when he had sex without a condom, he took a bath afterwards, which protected him from STIs and HIV. Three of the participants believed that condoms should not be used when breaking the girl’s virginity. The individual views of the three participants are shown as follows:

“The only time I did not use condoms is when I was breaking her virginity. After that, I was always using it until we broke up.”

Participant 13 said: “What I told myself is that at least let me have a first baby then afterwards, maybe I can use contraceptives because I cannot prevent without having any baby. I do not have a baby. That is a bit risky for my health and my womb.”

The participants’ concerns about the reliability and comfort of the condoms supplied by the MUT Clinic are very high, leading most study participants (16) to not use the supplied condoms. Because they are unemployed students, this means that they do not have money to buy condoms all the time. In summary, the results of the data collected indicated that students were inconsistent in their use of condoms.

### 3.3. Perceived Benefits of Contraceptives Use

The use of contraceptives is one of the most effective strategies for minimizing the risks linked with unwanted pregnancy. Generally, most of the students have a fairly good understanding of the use of contraceptives, but some of the students demonstrated some level of misinformation or lack of knowledge about contraceptives. The university clinic provides students with various types of contraception, except for IUCD and sterilization. Of the twelve female participants, eight reported using reliable contraceptives such as the pill (Triphasil or Nordette), injectable contraceptives (Petogen or Nur-Isterate), or the Implanon NXT. Only one participant used the intrauterine device (IUCD).

“Yes, I am on Nur-Isterate, and I am ok with it for almost 2 years now with no problems.”

Two participants had stopped using contraceptives altogether, and two participants had never used a contraceptive method. One participant was pregnant during the data collection. It was an unplanned pregnancy, and she decided to continue with the pregnancy.

Another participant indicated that contraceptive use could lead to infertility.

“What I told myself is that at least let me have a first baby then afterwards, maybe I can use contraceptives because I cannot prevent without having any baby. I do not have a baby. That is a bit risky for my health and my womb.”

Study participants from MUT expressed concern about contraceptive side effects that made them hesitant to use contraceptives. One participant stated that *“… you gain weight; people get pregnant while using contraceptives like Implant…”*.

Only one participant stated that he abstained from sex because he was not yet ready to become a father, indicating that he knew that abstinence was one of the methods to prevent unintended pregnancy. The perceived benefits influenced his change in behaviour.

The findings indicate that MUT students use the contraceptive services provided by MUT. However, the students have their preferred methods, which are sometimes determined by the side effects and convenience of a particular contraceptive method. If not addressed in a timely manner, these concerns may even lead young women to stop using contraceptives.

### 3.4. Alcohol and Drug Abuse

Most students (14 participants) reported drinking alcohol, and only three students were smokers. One participant smoked cannabis. The participant reported that alcohol hinders their judgment to practice safe sex. One participant stated that:

“There will be no condom use if both partners are drunk because they will not be thinking straight. They will want to do it now and their judgement will be disturbed.”

### 3.5. Pervieved Benefits of the Health Education Provided by the MUT

Health education helps students to acquire functional health knowledge, skills and beliefs needed to adopt and maintain healthy behaviours throughout their lives.

Four of the twenty participants indicated that they had not attended any of the health education events conducted by the MUT Clinic as part of health campaigns. Two of these participants stated that they did not know about the campaigns, while one said that it clashed with his academic classes or soccer practice. The other participant stated that he was familiar with the health information from his high school life orientation class and felt that the MUT Clinic health campaigns would provide him with the same information, which is why he did not attend.

One participant was asked if he attends the health talks provided by the MUT clinic staff during health campaigns, and his response was:

“I would not say yes because it’s things that I have learnt before from my life orientation classes in high school. It is just the reminders. I have not attended any health campaign; I knew about them, but it never came to me to attend them.”

Another respondent said that, as students, they did not listen:

“Yes, judging from what is happening, it is not working. You know us, students, we do not listen. Maybe it is the culture with us; maybe it is the times that we are growing in. If someone says, do not touch this fire, you will be burnt, we want to touch it and see if we will be burnt for real.”

Although MUT offers health education and information, condoms and contraceptives, there is still evidence of STIs and pregnancy rates. The health education and information provided by MUT should be improved to address and meet the needs of these students.

### 3.6. Developing a Greater Sense of Autonomy

Some of the study participants indicated that because of their young age and newfound freedom (autonomy) at university, they tend to engage in reckless behaviour away from home and their parents, leading to unplanned pregnancies, STIs, and HIV infections. The study found that the lack of parental support and supervision for MUT students away from home and parents plays an important role in students’ risky sexual behaviour. Five study participants cited the newfound freedom at the university as one of the reasons students experiment and become pregnant, contract STIs, or potentially contract HIV. One participant stated:

“I think they are being reckless, mostly at res; there are new students with newfound freedom where they can do whatever they want, whenever they want, then, a mistake happens, and they fall pregnant.”-

It was noted that age (maturity) and year of study play an important role in safer sex practices for MUT students.

One participant reported that:

“Mhhh…students at the university have different personalities and backgrounds and some of it we cannot prevent it. We come in different age groups with different levels of maturity. The judgement of a student who is coming straight out of high school is different from a person who left high school 2 or 3 years ago. The majority of students who find themselves in these mentioned problems are 1st-year students. The one who is straight from high school has newfound freedom and want to do everything now.”

### 3.7. Lack of Open Communication

Communication between people is crucial, especially between people who are in relationships, because they need to plan their lives, including their sex lives, to ensure their protection and safety. Four of the twenty participants indicated that they do not discuss their relationships and sex lives with their intimate partners. This was evident when they were asked if they knew what types of contraceptives their partners used. One participant (Participant one) stated that she does not talk about contraceptives in her relationship “I am not sure; we have not discussed such things.”

Participant eight stated that:

“Lack of communication between the partners. It is in us as Africans that we feel it is a shame to talk about sex even if we are in a relationship. We have sex, leave it in bed and talk about other stuff. Conversations about sex are rare between partners. It is what I have experienced as well. Maybe three out of ten couples discuss their HIV statuses before they even engage sexually.”

## 4. Discussion

In this study, we qualitatively explored the factors driving the high incidence of STIs and unplanned pregnancies among students at MUT, South Africa, despite the provision of condoms, contraceptives and health talks. The main themes that emerged from the interviews were multiple intimate partners, the perceived quality of condom use, the perceived benefits of contraceptive use, the perceived benefits of health education provided by the clinic staff at MUT Clinic, developing a greater sense of autonomy, lack of knowledge and misinformation about contraceptives, and alcohol and drug abuse. Although students have greater awareness and understanding of the risks linked with unprotected intercourse, most students do not see themselves personally at risk, and these findings are consistent with another study conducted in South Africa, which suggests that young people continue to engage in risky sexual behaviours despite having awareness of the risks of unplanned pregnancy and STI/ HIV infections [6].

We found that multiple partnerships and the lack of condom use was very high in this study. This is consistent with other studies [5,6,12]. Having multiple sexual partners is a known risk factor for HIV. Thus, reducing the number of sexual partners is an important strategy for HIV prevention. We noticed that male participants reported having two or more sexual partners compared to female participants. Other studies have found similar findings [6,8]. Male students engage continually in sexual activities despite their awareness of the risk linked with multiple sexual partners. We found that it is acceptable for males to have multiple sexual partners, and this was driven by peer pressure. Mthembu and co-authors reported similar findings, where peer pressure was a driver of risky sexual behaviour among university students in South Africa [6]. We noticed that only a few students reported not being sexually active, while only one student had never had sex. This finding may be attributed to personal, moral and religious influences. It is well known that religion gives a moral against sexual behaviours [13]. However, it is not certain if the students restrain from sexual activities due to religious principles.

In this study, the use of condoms remains inconsistent among the participants. The low and inconsistent use of condoms among the participants have great consequences among sexually active students, as it leads to the acquisition of HIV, STIs and unplanned pregnancies. Similarly, other studies have reported inconsistent and low condom use among university students [2,3,5,6]. Most students preferred condoms that they purchased themselves to those freely supplied by the MUT Clinic. The reason being that the MUT supplied condoms can easily break and they are of poor quality. The students did not trust the condoms provided by MUT; hence they were discouraged from using them. This highlights the lack of trust for the supplied condoms; thus this is likely to influence the decision to use condoms. These findings are in line with other studies that reported a negative perception of the government-sponsored condoms due to their poor quality, texture, smell and durability [6]. However, it is necessary to clear the doubt among the youth about the consistent use of condoms due to the dual benefits it offers, particularly in a high HIV setting.

The study found that young people develop a greater sense of autonomy as a result of their young age, and newfound freedom is also a factor influencing risky sexual behaviour among students. Young people are known to “try new things”, particularly when they away from their parents, families or relatives, which makes them vulnerable to risky sexual behaviour that leads to STIs, HIV infection and unplanned pregnancy. Additionally, the newfound freedom may also limit students’ capability to avoid peer pressure. Similarly, other studies have reported the same findings [14,15]. The findings from the study show that the lack of parental support and supervision for students influences risky behaviours among students.

We observed that some students attend the implemented health education programmes and some students do not. Studies shows that well-designed and well-implemented school-based health programmes can influence multiple health outcomes, including reducing sexual risk behaviours linked to HIV, STIs and unplanned pregnancies, thus reducing substance and tobacco use, and improving academic performance [15,16,17]. Health education programmes should be tailored to meet specific needs and interests for all university students, and these could help to reduce their risky behaviours.

Alcohol and drug use are known to increase risky sexual behaviour. Many studies have documented this relationship [5,6]. We found high use of alcohol among students in the study. Other studies have found high use of alcohol and drug use among university students [3,5,6]. Most participants who reported alcohol use were more likely in all cases to engage in risky sexual behaviour. Our observation is expected, as studies have shown that individuals engaging in HIV risk behaviours often engage in other high risk behaviours [3,6]. These findings suggest that more attention is needed on the role of alcohol in influencing sexual risk behaviours in the design and implementation of prevention programmes for university students.

Gender roles and norms are known as important drivers of the HIV epidemic [17,18]. Evidence exists that female relative disempowerment in relationships with their male counterpart reduced their ability to refuse sexual advances and negotiate safer sexual practices as well as condom use [18]. While for men, gender norms can aggravate concepts of masculinity that foster sexual prowess, virility and male control over females and frame condom use and fidelity as unmasculine [19,20,21]. In this study, only one female participant reported being able to negotiate safer sex practices. We did not gather enough information to assess gender norms in the study.

Communication channels such as the use of blogs, use of group SMS messages, student newspapers, students’ television and radio broadcasts, and students’ cell phone numbers should be registered to transmit prevention messages. Social media such as Facebook, which catches students’ attention, should be used to transmit prevention messages. All of these communication channels are possible approaches that should be taken into consideration when designing and implementing HIV- and STI-prevention interventions for university students. These channels should emphasize the importance of contraceptive use and reducing drugs and alcohol use and multiple sexual partners.

The use of a qualitative methodology alone limits the findings of the study since the study does not show any relationship via statistical results to determine the prevalence rate of risky sexual behaviours among the students. Similarly, the relationship between risky behaviours among first, second, third, fourth year and postgraduate students was not reported. Therefore, we recommend that future studies pertaining to risky sexual behaviours measure the incidence and occurrence of students’ risky sexual behaviours that lead to new HIV infections, STIs and unplanned pregnancies. In addition, the study was also limited to students residing in the university, hence the results of the study are only reflective of urban experiences of risky sexual behaviour among university students. Thus, generalizing the results to other settings should be done with caution.

## 5. Recommendations

The following recommendations resulted from the findings and were recommended by the participants and by the study:Institutional factors: Inadequate health education and information sharing opportunities. The study recommends that the MUT Clinic provide continuous health education in the clinic, which could be improved using audio-visual equipment such as televisions while students are waiting for their treatment. The MUT Clinic needs to refocus its health promotion programmes and involve health care recipients in the planning process to promote recipient acceptance. Currently, health education occurs during office hours, virtually via Microsoft Teams, and during health campaigns. In-person health education that took place in the residence hall was suspended in 2020 due to COVID-19 restrictions but has now been reinstated since November 2021.The MUT Clinic also needs to use MUT’s radio by booking slots as a platform for health education about STIs, HIV, and pregnancy and its complications. This will help shy students who are afraid of being seen at the clinic. The study participants suggest that the MUT Clinic should have its own Facebook page, different from the university’s main Facebook page, where it provides health information for students.Peer educators need to become more visible and promote health education, as study participants indicated that they are not visible in two students’ residences. Currently, there are no peer educators in some students’ residence, and they need to be made available.Socioeconomic and cultural factors: Personal responsibility challenge: The study recommends that students need to recognize their capacity for self-care (self-efficacy). It is the individual’s responsibility to practice safe sex by having a single sex partner and consistently using condoms to prevent the identified health problems and their complications.

## 6. Conclusions

This study suggests that there is a fairly good understanding of risky sexual behaviour and the perceived consequences among male and female students. Many participants were engaged in risky sexual behaviours that exposed them to HIV infection, STIs and unplanned pregnancies. The findings suggest that university students need multi-faceted interventions designed to address challenges with risky sexual behaviour including knowledge and benefits of condoms and contraceptive use in order to prevent STIs and unwanted pregnancies, and furthermore, providing psychosocial interventions to support these students with their autonomy.

## Data Availability

The qualitative datasets generated and analysed during the current study are not publicly available, but are available from the corresponding author upon reasonable request.

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
