# Peer review of "A Qualitative Exploration of Factors Influencing Non-Use of Sexual Reproductive Health Services among University Students in South Africa"

_ijerph, 2023, doi:10.3390/ijerph20032418_

Round 1

Reviewer 1 Report

I have attached a file with suggestions.  One last thing, consider that your audience is professionals at higher education institutions.  Rather than focusing on what is wrong with students, focus on what these results mean for professionals at education institutions.  If there is a perceived problem with quality of condoms so students won't use them, what can/ should a university do about that?  This paper needs to focus on how universities can change not how students need to change.  That was what authors stated was the purpose and also is how these results could be valuable to readers. 

Author Response

Reviewer’s comment 1:  No design was specified

Action: Thank you for the feedback. We have revised the study design section-we applied a qualitative descriptive research design. Please see lines 84.

Comment: Line 85: Were health sessions conducted on Teams or in person? Why were the interviews conducted via Teams?

Action: Thank you for this feedback. We have added -The health awareness sessions were conducted from June 14-25, 2021, via Teams. These sessions included all faculties within the University. One on one interviews were done from July to September 2021 via Teams as well, due to COVID-19 restrictions and protocols.

Comment: Line 86 & 94 contradict each other. Were first year students at the sessions? If so, how was their exclusion communicated to them? Also line 98 – students from different majors were eligible to participate but it doesn’t sound like you recruited by major – the students at the health sessions were invited to participate and the ones who were interested responded right?

Action: Thank you for this feedback. We have rephrased to state that all 589 students, from first to third year, including postgraduate students, were afforded equal opportunity to participate in the study. Please see line 86

Comment: Line 120 Male and female are not all genders. Considering you recruited students from the health sessions, your inclusion criteria were second, third- and fourth-year students of MUT who attended the health sessions in June 2021. That seems to cover the inclusion criteria right?

Action: Thank you for this feedback. We have added “and others” to the gender selection to reflect all genders. In addition, we have added the second, third, fourth year and postgraduate students to the inclusion criteria list.

Comment: Line 103 – How was the focus limited to heterosexual relationships? Were participants informed of that at the beginning of the interview? Were questions used that specified only heterosexual relationships? Were students involved in non-heterosexual relationships excluded? And why were only heterosexual relationships included?

Response: Thanks for this comment. We have rephrased to clearly state that all genders (male, female and other), all sexuality type were given equal opportunity to participate in the study. no participant was excluded due to their sexual orientation.

Comment: Lines 100- 106 – delete, this content has been conveyed in the intro. Line 103 needs more explanation and then to be moved. If you imposed this limitation from the beginning and it was incorporated in the interview guide, than explain that in the section about the interview guide.

If this focus was not imposed but, in the results, it was found that students only spoke about heterosexual relationships, than put that in results. Anoint

Inclusion of only heterosexual relationships whether imposed or how the results turned out, is a limitation so that should be identified as a limitation.

Action: Thanks for this feedback. We have deleted accordingly.

Comment: Line 145 – were these individual interviews? This line introduces the word discussion which brings confusion to the reader because now I am wondering if data were collected via focus groups. Line 144 included the word dialogues, which is a third term used to describe how data were collected. Changing terms is confusing to the reader. I recommend stating the interviews were with individual students

Action: Thanks for this feedback. Yes, they were one on one i.e. the researcher and the participant per interview session. We have deleted accordingly.

Comment: Line 172 – the quotes don’t support that participants had multiple partners to impress their friend groups. If anything, it seems like they have multiple partners because they want/ like to.

Action: Thanks for this feedback. We have deleted the suggestion that participants had multiple partners to impress their social group.

Comment: Lines 218-220 – beyond your data, delete

Action: Thanks for this feedback. We have deleted the “inconsistent use of condoms”.

Comment: Lines 282-285 – not related to your topic, delete

Action: Thanks for this feedback. We have deleted it accordingly.

Comment: Lines 291-296 – this interpretation would go in the discussion however I don’t think that results support this content anyways. They are suspicious of the quality of the condoms provided by MUT. Could that be something with merit? Are the condoms from the school different than what is in the store? Also the women reported getting contraceptives from MUT and they didn’t seem dissatisfied?

Action: Thanks for this feedback. We have deleted, and rephrased to state “The findings indicate that MUT students use contraceptive services provided by MUT. However, the students have their preferred methods, which are sometimes determined by the side effects and convenience of a particular contraceptive method. If not addressed in a timely manner, these concerns may even lead young women to stop using contraceptives.”-see line 295-299

Comment: Lines 304- 313 – there are no results reported in this section. Delete the section. Also, I think the women in this study have sex because they want to and choose to use birth control, two things that would indicate that they are not following gender norms of passivity. If there was content in which whty spoke about not using condoms because of pressure from male partner, that might support gender role as a barrier however that content was not presented in the table or results.

Action: Thanks for this feedback. We have deleted the section.

Comment: Lines 315-320 – this type of content doesn’t belong in results. Could be moved to discussion or deleted.

The heading for this section is not supported by the results. There is no information about the content of health education provided by MUT. Participants did not report that they don’t practice what was taught. If anything, participants’ comments indicate that the MUT education doesn’t meet their needs.

Action: Thanks for this feedback. We have rephrased to state “relevance of the health education provided by MUT. Please see line 311.

Comment: Lines 339-340 – consider interpreting the current state as perhaps that the institution could/ should do more. Rather than blaming students because these lines read as if MUT is doing all it can to prevent STI and pregnancy but still it happens, must be something wrong with the students. Like I said, it might be that what MUT is providing is not meeting the students’ needs and/ or there is only so much MUT can do. Really before you conclude that your services are not working, can you look at some data. Is there any data to show how many students use the services or STI/ pregnancy rates before or after services were implemented?

Action: Thanks for this feedback. We have revised the statement accordingly. Please see line 332

Comment: The section title Autonomy is judgemental. Yes I can see that a participant used the word “reckless” however, this section reads as if researchers are viewing that having sex is reckless. Or that having multiple partners is reckless. People have sex, its normal. I don’t think it will help students to label their behaviors negatively. Rather than judgement, the behaviors could be labeled with what is happening.

Action: Thanks for this feedback. We have changed the title to reflect-developing a greater sense of autonomy.

Comment: Lines 389-391 should be deleted. It is very judgmental. People have sex, that is a usual part of life. When support or authority figures portray judgemental attitudes, than young people turn away. Rather than presenting an attitude like,’ if only you would do what we tell you, than you’d be better off’ or ‘here at the university, we will provide everything you need to be safe, and if you don’t take advantage of what we have to offer, then something is wrong with you.’ I know that this is not what the authors meant, but this is how it sounds to me as a reader.

Action: Thanks for this feedback. We have rephrased to remove the judgemental nature of the statement.-“In this study, we qualitatively explored the factors driving high incidence of STIs and unplanned pregnancies among students at MUT, South Africa despite the provision of condoms, contraceptives and health talks.” Please see line 413

Comment: The discussion can be edited to correspond to the revised results. Be careful with language that sounds like you are blaming and judging young people. Its time to accept that young people have sex and for supportive adults to figure out ways to meet students’ needs in ways that they prefer. The recommendations at the end of the paper are student-centered and I like them.

Action: thanks for this feedback. We have revised the discussion section accordingly.

Reviewer 2 Report

The authors present a well designed qualitative study that is presented in a clear and easy to follow manner. My comments are minor and included below:

Methods:

-Interviews were transcribed by the researcher conducting the interview. Were these cross-checked by another independent transcriber?

-The authors state all gender identities were included. However, there is no mention of trans or non binary people. 

Results

-The authors reference first year students when commenting on the relationship between autonomy and sexual risk behavior. However, first year students were excluded from the study. Authors should clarify the relationship specifically for second, third and fourth year students.

-Table A1 would be better placed within the manuscript as Table 1. 

Author Response

The authors present a well designed qualitative study that is presented in a clear and easy to follow manner. My comments are minor and included below:

Methods:

Comment: -Interviews were transcribed by the researcher conducting the interview. Were these cross-checked by another independent transcriber?

Action: Thanks for this feedback. We have clarified and added this statement-“ Two researchers coded and transcribed the data from the in-depth-interviews and was analysed using an iterative process.” Please see line 137

Comment: -The authors state all gender identities were included. However, there is no mention of trans or non binary people. 

Action: Thanks for this feedback. We have revised the statement to include “other” genders as a category of trans or non binary people.

Results

-The authors reference first year students when commenting on the relationship between autonomy and sexual risk behavior. However, first year students were excluded from the study. Authors should clarify the relationship specifically for second, third and fourth year students.

Action: Thanks for this feedback. We have revised the statement accordingly. Please see line 343.

Response: The study did not explore the relationships specifically for second, third, and fourth year students. However, this is an important feedback and we have made mention of it in the limitation section. Please see line 518.

Commet: -Table A1 would be better placed within the manuscript as Table 1. 

Action: Thanks for this comment. We have added the Table within the manuscript

Round 2

Reviewer 1 Report

Line 318 - I wouldn't use the word weed, too colloquial. 

Line 334 - I like this heading much better, it also is reflective of the data. 

I like the authors' revisions.  Each section is clear and I understand what the study was about, how it was conducted and can see how the results and discussion are linked to the data.  

I think the tone is better too, not judgmental of the students, just reporting the situation as it is. 

I really like the recommendations.  I am going to share those ideas with my university too.  I am at a catholic university with restrictions on even talking about contraception and I impressed with the range of services offered at MUT.  Once your article is published, I will use it in my work at my university to improve sexual health for our students. 

Needs minor editing throughout for such things as tense (uses present form when the sentence implies past) and singular/ plural agreement for example lines 495-497 should be students' not student's. 

Author Response

Line 318 - I wouldn't use the word weed, too colloquial. 

Action: Thank you for this feedback. We have corrected according to read “ Cannabis”

Line 334 - I like this heading much better, it also is reflective of the data. 

Response: Thank you

I like the authors' revisions.  Each section is clear and I understand what the study was about, how it was conducted and can see how the results and discussion are linked to the data.  

I think the tone is better too, not judgmental of the students, just reporting the situation as it is. 

I really like the recommendations.  I am going to share those ideas with my university too.  I am at a catholic university with restrictions on even talking about contraception and I impressed with the range of services offered at MUT.  Once your article is published, I will use it in my work at my university to improve sexual health for our students. 

Needs minor editing throughout for such things as tense (uses present form when the sentence implies past) and singular/ plural agreement for example lines 495-497 should be students' not student's. 

Action: Thank you for this feedback. We have edited accordingly. Please see edits in track changes